# The experiences of adult heart, lung, and heart-lung transplantation recipients: A systematic review of qualitative research evidence

Claire Stubber[º], Maggie Kirkman[iD]*[º]

Global and Women's Health, Public Health and Preventive Medicine, Monash University, Melbourne, Victoria, Australia

[º] These authors contributed equally to this work.
* maggie.kirkman@monash.edu

**Data Availability Statement:** All relevant data are within the paper and its Supporting Information files.

## Abstract

### Aim

To review evidence about the experience of being the recipient of a donated heart, lungs, or heart and lungs.

### Design

A systematic review (registered with PROSPERO: CRD42017067218), in accordance with PRISMA guidelines.

### Data sources

Seven databases and Google Scholar were searched in May 2017 and July 2019 for papers reporting English-language research that had used qualitative methods to investigate experiences of adult recipients.

### Review methods

Quality was assessed and results were analysed thematically.

### Results

24 papers (reporting 20 studies) were eligible and included. Their results were organised into three chronological periods: pre-transplant (encompassing the themes of 'dynamic psychosocial impact', 'resources and support'), transplant ('The Call', 'intensive care unit'), and post-transplant ('dynamic psychosocial impact', 'management', 'rejection'). Sub-themes were also identified. It was evident that contemplating and accepting listing for transplantation entailed or amplified realisation of the precipitating illness's existential threat. The period surrounding transplantation surgery was marked by profound, often surreal, experiences. Thereafter, although life usually improved, it incorporated unforeseen challenges. The transplantation clinic remained important to the recipient. The meaning of the clinic and its staff

**Funding:** The authors were supported by a grant from the Grenet Foundation courtesy of Ann Hyams. The funders had no role in study design, data collection and analysis, decision to publish, or preparation of the manuscript.

**Competing interests:** The authors have declared that no competing interests exist.

could be both reassuring (providing care and support) and threatening (representing onerous medical requirements and potential organ rejection).

## Conclusion

This review has implications for the psychosocial care of transplant recipients and indicates the need for further research to gain insight into the experience of receiving a donated heart and/or lung.

## Impact

Medical consequences of heart and lung transplantation are well documented; this is the first systematic review of research using qualitative methods to investigate the experience of heart, lung, and heart-and-lung transplantation. The psychosocial impact of transplantation was found to be dynamic and complex, with notable features evident before, during, and after transplantation. Clinic staff remained significant to recipients. It is clear that recipients need continuing psychosocial as well as medical support.

## Introduction

Heart, lung, and heart-lung transplantation is now standard clinical treatment for some intractable heart failure [1] and end-stage pulmonary diseases [2]. There is no single register that records every incidence of heart, lung, or heart-lung transplantation. The two most prominent registers are published by the International Society for Heart Lung Transplantation and the Global Observatory on Donation and Transplantation. For the 12 months from 1 July 2016–30 June 2017 there were 4,547 adult heart transplants [3], 4,095 adult lung transplants, and 47 adult heart-lung transplants [4] reported to the International Society for Heart Lung Transplantation. The Global Observatory on Donation and Transplantation, a collaboration between the World Health Organization and the Spanish Transplant Organization, estimated that there were 6,865 heart transplants and 5,500 lung transplants, globally, in the year 2018 (www.transplant-observatory.org/who-ont).

It is difficult to establish for any given year how many people worldwide are on waiting lists for a donor heart, lung, or heart and lung: Constant fluctuation is caused by people moving on and off lists because of death, transplantation, and variation in health status. Worldwide in 2017, there were 16,607 people active at some time on a waiting list for heart transplantation and 9,373 people active at some time on a waiting list for lung transplantation [5].

Transplantation of hearts and lungs is problematic, both for the medical institution offering the procedures and for the recipients. Organ procurement and allocation are the main institutional difficulties [6, 7]. Organ transplantation recipients confront diverse challenges, only some of which become apparent when organ transplantation is presented as an option. The obligation of informed consent requires patients to decide what is best for their own health; in the case of cardiothoracic organ transplantation, patients must determine whether a particular procedure will save their life, but what, if any, improvements they can expect in their quality of life [8]. Ideally, a patient will be guided by a clinician who can explain the benefits and disadvantages of such treatment and make a recommendation guided by the patient's best interests. There are many things patients may consider when they are asked if they want to join an organ transplant waiting list [9], some of which are made explicit while others are implicit;

some may become apparent only long after transplantation and some considerations may never be relevant to a particular patient [10]. The decision to accept a place on the waiting list can therefore be an exercise in imagination [11].

## Background

It has been known for decades that people on the waiting list are subject to numerous stressors [12], including deterioration in physical health, isolation, stigma associated with the receipt of donated organs (such as comparisons with Frankenstein's monster and a perception of complicity in the receipt of cadaveric organs) [13], perhaps relocation to the city in which the transplant will take place, fear of death, anxiety about whether their pager or phone would alert them when an organ became available, and the effects of false alarms [14, 15]. Post-transplantation, there can be adverse psychological, psychosocial, and medical consequences, including the onset of diabetes, hypertension, renal insufficiency, osteoporosis, diverse malignancies, distressing changes to appearance, and opportunistic infections [16–18]. Heart and lung transplant recipients know that they are vulnerable to acute and chronic organ rejection by the immune system [19]. Chronic rejection in heart transplant recipients (cardiac allograft vasculopathy) accounts for 30% of post-transplantation deaths; chronic rejection in lung transplant recipients (bronchiolitis obliterans syndrome) is the leading cause of death for those who survive a year post-transplantation [20, 21]. Recipients can experience guilt [22] arising from the ethical questions intrinsic to organ donation about harm (whether being a recipient harms the donor and their family), beneficence (whether any good arising from the donation outweighs any harm), equity (of access to donated organs and life-sustaining medical treatment), justice (whether the recipient is deserving of the organ), and utility (whether transplanting the organ promotes wellbeing) [23]. A review of quantitative psychological studies found improvements in recipients' mental health and health-related quality of life after lung transplantation [24]. A review of seven qualitative studies of recipients of donated hearts identified the importance of social support, especially in promoting a sense of agency [25]. There has not been a review of qualitative research on recipients of donated hearts and/or lungs; it is therefore time to update the Conway et al. [25] review and to extend it to recipients of donated lungs.

## The review

### Aim

The aim of this review was to assemble evidence about the meaning of heart, lung, or heart and lung transplantation to adult transplantation recipients.

### Design

A systematic review was conducted in accordance with PRISMA guidelines [26] of papers reporting research that had used qualitative research methods to investigate recipients' perspectives on heart, lung, or heart and lung transplantation. The review protocol was registered on PROSPERO (CRD42017067218). As a review of published work, ethical approval was not required.

### Search methods

Papers were eligible for inclusion in the review if they reported original research using qualitative research methods to investigate adult recipients' experiences of heart, lung, or heart and lung transplantation and were published in English in peer-reviewed journals. No date limits were set. Exclusion criteria were that others (such as parents or support persons) described

patients' experiences, that no participants had yet received a transplanted organ, and that the researchers reported transplant recipients' views only on a specific intervention.

Seven databases (Ovid, Ebscohost, ProQuest, Web of Science, Family and Society Plus, Sociological Abstracts, and International Bibliography of the Social Sciences) were individually searched using the MESH terms ['transplant*'] AND ['heart' OR 'lung' OR 'heart-lung' OR 'heart and lung' OR 'cardiothoracic'] AND ['qualitative' OR 'interviews' OR 'experience']. To ensure that eligible papers not found on these databases were detected, we searched Google Scholar using the terms 'heart transplant qualitative' and 'lung transplant qualitative' and examined the reference lists of articles identified from the database search. The initial search was conducted in May 2017 with a second search for any subsequent publications on 22 July 2019.

## Search outcome

The search and selection process is detailed in Fig 1.

The 24 eligible papers (reporting data from 20 studies) are summarised in Table 1.

The research was conducted in 11 countries, all categorised as high or upper-middle income: Australia (1), Brazil (1), Canada (2 studies, 3 papers), Denmark (1), Iran (1), Scotland (1 study, 2 papers), Sweden (5 studies, 7 papers), Switzerland (1), Spain (1), UK (3), and USA (3). All used interviews (in-depth or semi-structured) to gather data. The majority (278/353: 79%) of recipients had received a heart; 68 had received a lung or lungs; and one had received a heart and lungs. Most participants were recruited from their post-transplantation clinics. There were 353 participants (aged 16–72 years) in the studies, with slightly more female participants than male participants. Few other participant characteristics (such as socio-economic status, sexual orientation, and ethnic identity) were presented. With the exception of a paper specifically reporting women's experiences [27], no paper reported gendered aspects of the experience of transplantation.

## Quality appraisal

The quality of selected articles was assessed using an established checklist [28] modified by the inclusion of an additional criterion that we consider to be essential: the presence of a statement

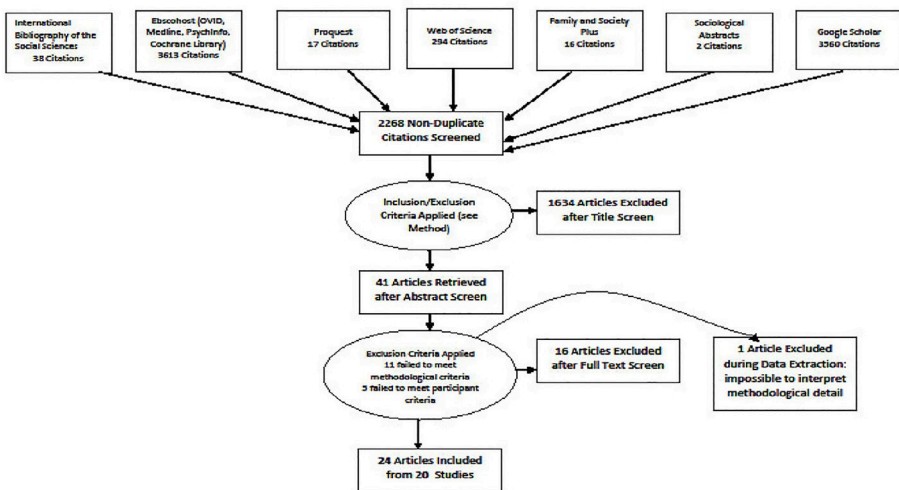

**Fig 1. Flow chart: Articles yielded by the search, process of exclusion, & articles (+ studies) reviewed.**

**Table 1. Summary of reviewed papers.**

| Author (date), country | Aim | N participants, age range in years, time since transplant | Data collection | Analysis | Themes [†] |
|---|---|---|---|---|---|
| Ålmgren et al. (2017a), Sweden [‡] | "in-depth exploration of the meaning of uncertainty during the first year after a heart transplantation" | 14 heart recipients (4 women, 10 men) aged 28–67, 1 year post transplant | In-depth interviews | "phenomenological-hermeneutic", "thematic structural" | Expectations Inadequacy (Post) Medical Reject |
| Ålmgren et al. (2017b), Sweden [‡] | "in-depth exploration of self-efficacy among heart transplant recipients by means of Bandura's self-efficacy theory" | 14 heart recipients (4 women, 10 men) aged 28–67, 1 year post transplant | In-depth interviews | "Directed content" | Resource (Pre) ICU Hope Donor Internal (Post) External (Post) |
| Dabbs et al. (2004), USA | "to explore how lung recipients perceive, interpret, and relate symptoms to the threat of rejection" | 14 lung recipients (7 women, 7 men) aged 28–69, 27 days-9 years post-transplant | In-depth interviews (1 in person + at least 1 by phone) | "Consistent with the grounded theory approach" | Hope Expectations Inadequacy (Post) Medical Reject |
| Evangelista et al. (2003), USA | "to explore women's psychological recoveries from heart transplant surgeries" | 33 female heart recipients, mean age 62.3, 1–22 years post-transplant | Semi-structured interviews | "Content" | Threats Inadequacy (Pre) Resource (Pre) Hope Expectations Medical Internal Reject |
| Flynn et al. (2014), UK | "to explore the narratives of people who have had a heart or lung transplant and who report experiencing delirium in the ICU" | 11 heart or lung recipients (4 women, 7 men) aged 40–69, 6.5 months-14 years post-transplant | Open-ended interview converted by researchers into a narrative | "Narrative" | Threats Inadequacy (Pre) The Call ICU Hope Expectations Inadequacy (Post) Medical Internal External Reject |
| Ivarsson et al. (2013a), Sweden [§] | "to illuminate how patients, six months after a heart or lung transplantation, experienced the information and support they received in connection with the transplantation" | 16 heart or lung recipients (9 women, 7 men) aged 16–67, 6 months post-transplantation | Semi-structured interviews | "Qualitative content" | Threats Inadequacy (Pre) ICU Expectations Donor Inadequacy (Post) Medical External Reject |

*(Continued)*

**Table 1.** (Continued)

| Author (date), country | Aim | N participants, age range in years, time since transplant | Data collection | Analysis | Themes [†] |
|---|---|---|---|---|---|
| Ivarsson et al. (2013b), Sweden [§] | "to describe the patients' retrospective experiences of the information and support they received while on the heart or lung transplant waiting list" | 16 heart or lung recipients (9 women, 7 men) aged 16–67, 6 months post-transplantation | Semi-structured interviews | "Qualitative content" | Threats Inadequacy (Pre) Resource (Pre) The Call |
| Kaba et al. (2000), Scotland [¶] | "to explore the coping strategies of heart transplant recipients with the intention of identifying helping strategies for cardiac nurses" | 42 heart recipients (7 women, 35 men) aged 32–61, 2–24 months post transplant | In-depth interviews | "Constant comparative method" | Threats Inadequacy (Pre) Resource (Pre) Donor Medical Internal |
| Kaba et al. (2005), Scotland [¶] | "to explore psychological problems experienced by heart transplant recipients" | 42 heart recipients (7 women, 35 men) aged 32–61, 2–24 months post transplant. | In-depth interviews | "Constant comparison" | Donor Inadequacy |
| Lawrence et al. (2008), USA | "to (1) confirm the relationship between maturity, self-concept, and adherence found in the first study and (2) develop themes of interpersonal relationships with family and friends among adolescent and young adult transplant recipients" | 46 heart recipients aged 15–31, 11–18 years post-transplant. Sex not stated | Semi-structured interviews | Not named, but description equates with thematic (within "grounded theory") | Medical External Internal |
| Lundmark et al. (2016), Sweden | "to develop the concept analysis by Allvin et al. (2007) from lung recipients' perspective of their post-transplant recovery process and to identify the recovery trajectories including critical junctions in the post-transplant recovery process after lung transplantation" | 15 lung recipients (1 had also received a heart) (5 women, 10 men) aged 26–70, 1 year post transplant | Open-ended interviews | "Directed concept" | Hope Expectations Inadequacy (Post) Guilt Medical External (Post) |
| Macdonald (2006), UK | "to examine the lived experience of patients with CF . . . and of coping with the rigours of chronic illness while waiting for a lung transplant" | 4 male lung candidates and 4 lung recipients (3 women, 1 man) aged 19–40, up to 3 years post-transplant | Semi-structured interviews | "Content" | Threats Inadequacy (Pre) Resource (Pre) The Call Expectations Inadequacy (Post) |
| Mauthner et al. (2015), Canada [#] | "to study transplant recipients' experiences of incorporating a transplanted heart" | 25 heart recipients (7 women, 18 men) aged 18–72, 1–10 years post transplant | Semi-structured interviews | "Visual methodology", "themes" | Donor Identity |
| Moloney et al. (2007), Canada | "To identify, from the perspective of patient, the information received and desired on transplantation to make an informed decision; the actual and preferred ways of receiving information; and the involvement of support persons in the decision" | 8 lung candidates (5 women, 3 men) and 14 recipients (6 women, 8 men) aged 22–65, up to 7 years post-transplant | Semi-structured interviews | "Qualitative content" | Resource (Pre) Expectations Internal |

(*Continued*)

**Table 1.** (Continued)

| Author (date), country | Aim | N participants, age range in years, time since transplant | Data collection | Analysis | Themes [†] |
|---|---|---|---|---|---|
| Neukom et al. (2012), Switzerland | To answer: "how is the relationship between recipient and donor structured in the narratives? Do these empirical findings lend themselves to psychoanalytic theories of the psychic integration of transplanted organs?" | 6 lung recipients (3 women, 3 men), at least 12 months post-transplant | "Semi-standardised" interviews | "JAKOB narrative" | Donor Inadequacy (Post) |
| Nilsson et al. (2008), Sweden | "To investigate perceptions of graft rejection and different methods to obtain knowledge about graft rejection among adult organ transplant recipients" | 4 heart (1 woman, 3 men), 4 lung recipients (3 women, 1 man) (+4 kidney, 4 liver) aged 26–58, 6 months-9 years post-transplant | In-depth interviews | "Constant comparison" | Inadequacy (Post) Medical Internal Reject |
| O'Brien et al. (2014), Australia | "to explore the lived experience of successful heart transplantation, particularly how heart recipients experience and manage 'the tyranny of the gift'" | 13 heart recipients (5 women, 8 men) aged 35–72, 10 weeks-11 years post transplant | Semi-structured & brief follow-up interviews | "Interpretative phenomenological" | Threats Inadequacy (Pre) Hope Inadequacy (Post) Medical External |
| Palmar-Santos et al. (2019), Spain | "To explore the experiences of patients after receiving a heart from a donor" | 12 heart recipients (6 women, 6 men) aged 27–70, 3 months-10 years post transplant | In-depth interviews | "Discourse", then "themes and sub-themes" | Threats Inadequacy (Pre) The Call ICU Donor Medical |
| Peyrovi et al. (2014), Iran | "to explore and gain deep insights about living with a heart transplant" | 11 heart recipients (2 women, 9 men) aged 21–55, 7 months-18 years post-transplant | In-depth interviews | "Diekelmann's hermeneutical" | Hope Expectations Donor Inadequacy (Post) Medical External |
| Poole et al (2016), Canada [#] | "to examine the loss and grief experiences of patients waiting for and living with new hearts" | 15 heart recipients aged 18–72. Sex not stated, about 2–6 years post-transplant | Secondary analysis of existing data | "a qualitative visual method" using NVivo; appears to be consistent with thematic | Inadequacy (Pre) Donor Inadequacy (Post) Guilt |
| Sadala & Stolf (2008), Brazil | "to investigate the HT experience by choosing a qualitative method aimed at describing the meanings patients give to the experience they lived" | 26 heart recipients (6 women, 20 men) aged 17–71, 4–17 years post-transplant | In-depth interviews | "Phenomenological" | Threats Inadequacy (Pre) Hope Inadequacy (Post) Medical External (Post) Reject |
| Sanner (2003), Sweden | "to examine how organ recipients in late modernity conceived the special features that distinguish the transplantation from other treatments, namely that vital, 'living' organs are transferred from one human being (deceased or living) to another" | 15 heart recipients (5 women, 10 men) (+ 23 kidney) aged 30–64, 1–3 weeks post-transplant, repeated up to 2 years post-transplant | 1–5 "open" interviews | "Qualitative . . . on three themes" | Inadequacy (Pre) ICU Donor Identity Inadequacy (Post) Reject |

(*Continued*)

**Table 1.** (Continued)

| Author (date), country | Aim | N participants, age range in years, time since transplant | Data collection | Analysis | Themes [†] |
|---|---|---|---|---|---|
| Thomsen & Jensen (2009), Denmark | "investigating the experiences of everyday life after lung transplantation of patients with previous COPD" | 10 lung recipients (5 women, 5 men) aged 51–69, 7 months-7 years post-transplant | Semi-structured interviews | "Qualitative content" | Inadequacy (Pre) Hope Medical Reject |
| Waldron et al. (2017), UK | "to explore the experience of heart transplant in young adults" | 9 heart recipients (4 women, 5 men) (age at interview not given), 7 months-9.5 years post-transplant | Semi-structured interviews | "Interpretative phenomenological" | Threats Inadequacy (Pre) Resource (Pre) Expectations Donor Inadequacy (Post) Internal (Post) |

[†] See Table 3 for details of themes

[‡] Same study (identified from identical HREC number & participant information)

[§] Same study

[¶] Same study

[#] Same study

of approval from an institutional human research ethics committee. This addition meant that scores (ranging from 0 to 1) were based on 11 items rather than 10. Authors scored independently and resolved any differences by discussion; final scores were reached by agreement.

Quality assessment scores ranged from 0.77 to the maximum of 1.0; details are in Table 2. Most papers (17) scored zero for reflexivity. There was no statement of approval from a human research ethics committee in two papers [29, 30]; authors' responses to our written queries are noted in Table 2.

## Data abstraction

Both authors collaborated on data abstraction and synthesis, which benefited from the insights of Author 1 as a recipient (in 1996) of a donated heart and lungs. Some papers required detailed searching to identify the data of interest because they were not presented systematically or succinctly. Where eligible studies included ineligible participants (such as family members or recipients of other organs) we excluded their data.

Data abstracted were the country in which the research was conducted; the aim; the number of participants, the organ(s) received, their sex, age range, and time since transplant; the method of data collection; the method of analysis; and details of the Results or Results and Discussion sections.

## Synthesis

Abstracted results were analysed thematically, using a standard, iterative, qualitative method [31]. As new themes were identified in each paper, all papers were searched to establish whether that theme could be found there. Whether or not the reviewed papers presented their data thematically, we generated our own themes from the results and took care not to privilege

Table 2. Quality assessment (after Kmet et al., 2004).

| Author (Date) | Question /objective clearly stated? | Design evident and appropriate to answer study question? | Ethics approval? [†] | Context for study is clear? | Connection to a theoretical framework/ wider body of knowledge? | Sampling strategy described, relevant and justified? | Data collection methods clearly described and systematic? | Data analysis clearly described, complete and systematic? | Use of verification procedure(s) to establish credibility of the study? | Conclusion supported by the results? | Reflexivity of the account? | Score |
|---|---|---|---|---|---|---|---|---|---|---|---|---|
| Älmgren et al. (2017) | Yes | Yes | Yes | Yes | Yes | Yes | Yes | Yes | Yes | Yes | Partial | .95 |
| Älmgren et al. (2016) | Yes | Yes | Yes | Yes | Yes | Yes | Yes | Yes | Yes | Yes | Partial | .95 |
| Dabbs et al. (2004) | Yes | Yes | No[‡] | Yes | Yes | Yes | Yes | Partial | Yes | Yes | No | .81 |
| Evangelista et al. (2003) | Yes | Yes | Yes | Yes | Yes | Yes | Yes | Yes | Yes | Yes | Partial | .95 |
| Flynn et al. (2014) | Yes | Yes | Yes | Yes | Yes | Yes | Yes | Yes | Yes | Yes | Yes | 1 |
| Ivarsson et al. (2013a) | Yes | Yes | Yes | Yes | Yes | Yes | Yes | Yes | Yes | Yes | Yes | 1 |
| Ivarsson et al (2013b) | Yes | Yes | Yes | Yes | Yes | Yes | Yes | Yes | Yes | Yes | No | .90 |
| Kaba et al. (2000) | Yes | Yes | Yes | Yes | Yes | Yes | Yes | Yes | Yes | Yes | No | .90 |
| Kaba et al. (2005) | Yes | Yes | Yes | Partial[§] | Yes | Partial[§] | Partial[§] | Yes | Yes | Yes | No | .77 |
| Lawrence et al. (2008) | Yes | Yes | Yes | Partial[¶] | Yes | Yes | Yes | Partial | Yes | Yes | No | .86 |
| Lundmark et al (2016) | Yes | Yes | Yes | Yes | Yes | Yes | Yes | Yes | Yes | Yes | No | .90 |
| Macdonald (2006) | Yes | Yes | Yes | Yes | Yes | Yes | Yes | Yes | Yes | Yes | Partial | .95 |
| Mauthner et al. (2015) | Yes | Yes | Yes | Yes | Yes | Yes | Yes | Yes | Yes | Yes | No | .90 |
| Moloney et al. (2007) | Yes | Yes | Yes | Yes | Yes | Yes | Yes | Yes | Yes | Yes | No | .90 |
| Neukom et al. (2012) | Yes | Yes | No[#] | Yes | Yes | Yes | Yes | Yes | Yes | Yes | No | .81 |
| Nilsson et al. (2008) | Yes | Yes | Yes | Yes | Yes | Yes | Partial | Partial | Yes | Yes | No | .86 |
| O'Brien et al. (2014) | Yes | Yes | Yes | Yes | Yes | Yes | Yes | Yes | Yes | Yes | No | .90 |
| Palmar-Santos et al. (2019) | Yes | Yes | Yes | Yes | Yes | Yes | Yes | Partial | Yes | Yes | No | .86 |
| Peyrovi et al. (2014) | Yes | Yes | Yes | Yes | Yes | Yes | Yes | Yes | Yes | Yes | No | .90 |

(Continued)

Table 2. (Continued)

| Author (Date) | Question /objective clearly stated? | Design evident and appropriate to answer study question? | Ethics approval? † | Context for study is clear? | Connection to a theoretical framework/ wider body of knowledge? | Sampling strategy described, relevant and justified? | Data collection methods clearly described and systematic? | Data analysis clearly described, complete and systematic? | Use of verification procedure(s) to establish credibility of the study? | Conclusion supported by the results? | Reflexivity of the account? | Score |
|---|---|---|---|---|---|---|---|---|---|---|---|---|
| Poole et al. (2016) | Yes | Yes | Yes | Partial^Δ | Yes | Yes | Yes | Yes | Yes | Yes | No | .95 |
| Sadala & Stolf (2008) | Yes | Yes | Yes | Yes | Yes | Partial | Yes | Yes | No | Yes | No | .77 |
| Sanner (2003) | Yes | Yes | Yes | Yes | Yes | Yes | Yes | Yes | Yes | Yes | No | .90 |
| Thomsen & Jensen (2009) | Yes | Yes | Yes | Yes | Yes | Yes | Yes | Yes | Yes | Yes | No | .90 |
| Waldron et al. (2017) | Yes | Yes | Yes | Yes | Yes | Yes | Yes | Yes | Yes | Yes | Yes | 1.0 |

† Additional criterion

‡ Personal communication from first author (23 May 2019): "The study published in Soc Sci and Med was approved by the University of Pittsburgh IRB # 0110142"

§ Information in an earlier paper (also in this review)

¶ Information in an earlier paper (not in this review)

# Personal communication from first author (17 October 2017): "we obtained a written consent from each involved patient when we started our research. . . . it wasn't mandatory nor a standard procedure to apply for an approval from an official ethic committee in these days."

Δ No explicit statement of where data were gathered; Canada is implied.

research that had been analysed thematically. In developing the thematic scheme that best synthesised the results from all studies, diagrams and flow charts were used to aid conceptual understanding. No software was used in the analysis. Any differences of opinion between the authors were resolved by discussion. All aspects of the analysis were undertaken and completed by collaboration and discussion between the authors, who reached agreement on every detail.

## Results

The identified themes, all concerning psychosocial aspects and practicalities of organ transplantation, were most efficiently categorised chronologically: *Pre-transplant*, *Transplant*, and *Post-transplant*. Papers from seven studies reported results from all three periods [32–39]; the remainder were concerned with one or two periods only, such as the post-transplant experience [29, 30, 40–47] or the time surrounding transplantation [33, 37, 39, 48]. Each chronological period had several sub-themes (often repeated in more than one chronological period and often interconnected), which are described below. Themes and sub-themes are listed in Table 3.

### Pre-transplant

Participants were reported as reflecting on their pre-transplant lives, whether from the other side of organ transplantation or, in the case of three studies [37, 46, 49], from the perspective of those still awaiting an organ. We identified both the *Dynamic psychosocial impact* of contemplating organ transplantation and the mechanisms of *Resources and support*.

**Table 3. Identified themes and subthemes.**

| Chronological period Theme Subtheme | | Abbreviated form |
|---|---|---|
| **Pre-transplant** | **Dynamic psychosocial impact** | |
| | Threats to self | Threats |
| | Sense of inadequacy | Inadequacy (Pre) |
| | **Resources and Support** | |
| | Internal sources (optimism, positive thinking, faith, hope) | Internal (Pre) |
| | External sources (clinicians, information, peers) | External (Pre) |
| **Transplant** | The Call | The Call |
| | Intensive care unit | ICU |
| **Post-transplant** | **Dynamic psychosocial impact** | |
| | Hope | Hope |
| | Expectations and reality | Expectations |
| | Donor<br>Identity | Donor<br>Identity |
| | Sense of Inadequacy<br>Guilt | Inadequacy (Post)<br>Guilt |
| | **Management** | |
| | Medical<br>Support: | Medical |
| | *Internal sources* (faith, goals) | Internal (Post) |
| | *External sources* (allied health, nurses, workplaces, families, social groups, peers) | External (Post) |
| | **Rejection** | Rejection |

The psychosocial impact of being listed for a donated organ was profound and variable. We categorised research participants' reported feelings as *Threats to self* and a *Sense of inadequacy*. It was evident that contemplating and accepting listing for transplantation entailed or amplified realisation of the illness's existential threat. Participants in three studies described their shocked reactions to the prospect of transplantation and the anxiety provoked by eligibility tests [33, 37, 47]. Fear was presented as a common accompaniment to thoughts about transplantation and its implications, as well as to the chronic illness that necessitated a donated organ [27, 32–34, 38]. Potential recipients worried about the effects on their family; those who were parents expressed particular concern about when and how to discuss illness and transplantation with their children [27, 34, 35, 38, 50].

The clinic attended by those waiting for an organ was reported to be a central figure in some participants' accounts; this figure could be both menacing and reassuring [34, 37, 44, 48]. Because phone calls from the clinic could mean that an organ was available, any clinic call created anxiety. If the call was simply a welfare check, it reassured some participants that the clinic had not forgotten them but it disappointed others. "False alarms," where the patient had been called to hospital for a transplant that did not proceed, were reported by participants as provoking uncertainty, frustration, and disappointment [37, 47]. Clinic visits could incite the same feelings in those waiting [48].

Participants were reported as describing a sense of personal inadequacy engendered by the idea and process of transplantation [39, 47, 48, 51]. Guilt often underpinned this, whether from feeling responsible for the illness that necessitated transplantation or because of hoping for the death of a suitable organ donor [39, 47, 51]. It was found in one study that participants had been eager to prove that they deserved a place on the transplantation list [44]; this could be understood as revealing the absence of a sense of entitlement.

The limitations imposed by chronic illness and the need for a donated organ were reported in 10 papers from 8 studies to have left participants without a sense of agency; illness and waiting for "The Call" created an uncertain environment in which periodic crises replaced a familiar life [27, 32–35, 37, 47, 48, 50]. Participants in one study were reported as saying that a long wait for transplant allowed for "brooding and reflection" which led to "fear" [32]. Illness was also reported to have been experienced as isolating; at the same time, where life-saving equipment was needed, it made one inescapably dependent on others [34, 48]. Even when those on the waiting list consciously attempted to distance themselves from the severity of their illness, the demands of managing illness made this psychologically and physically difficult [48]. Researchers in two studies interpreted some of their participants' actions as attempts to regain control [37, 50]. Actions included avoiding anxiety-provoking stimuli, such as news of war, and being meticulous about diet, one of the few ways in which participants felt they could contribute to optimal pre-transplantation health.

In their attempts to cope with the existential threat and practicalities of their circumstances, research participants reported drawing on various *Resources and supports*; they also described the support they wished had been available. Some resources were internal: optimism, positive thinking, faith, and hope [27, 43, 50]. Clinicians constituted an important external source of support [27, 34, 37, 44, 49], both psychological and practical [27]; practical support could be as simple as information [34]. There were reports that research participants had expected or needed pre-transplant support that had not been forthcoming [34, 49]. Peer support could be valuable to those awaiting transplantation, but it was not inevitably so. Peers could provide reassurance, first-hand information, explanations of the decision-making process, inspiration, a sense of trust in the future, and hope [34, 37, 49, 50]. The accessibility of peer support varied, with Swedish participants reporting that they had met with transplant recipients [34] whereas Canadian participants did not have that option [49]. Participants in another study were said to have avoided peers, preferring not to know what lay ahead [50].

## Transplant

One study [33] focused on the transplantation operation and the post-operative period; other studies included these experiences. We identified *The Call* and the intensive care unit (*ICU*) as the major sub-themes of this relatively brief but intense time.

"*The Call*" is the telephone call summoning the candidate to hospital because an organ is available. When The Call does not result in surgery it is known as a false alarm. Research participants were reported as viewing The Call with a mixture of "fear", "disbelief," and "reverence"; it could be the culmination of waiting for salvation, "a very beautiful experience", or the instigator of "shock," "emptiness," and, in the researchers' words, "anxiety" and "uncertainty" [33, 34, 37, 38].

Being in the *ICU* after surgery generated powerful yet diverse memories. Some research participants reported experiencing physical well-being, euphoria, and relief in the ICU [32, 33, 38, 39]. A study designed to understand the phenomenon known as ICU delirium explored the vivid post-operative hallucinations often experienced by organ recipients, which can be extremely frightening [33]. Physical mobilisation while in the ICU could provoke anxiety, although once accomplished it boosted confidence [32]. Recipients were found to be grateful for information given before surgery about physical activity during their ICU stay and for support for such activities [32, 35].

## Post-transplant

The main focus of most of the reviewed papers was recipients' lives after transplantation, whether in general or with a specific focus: adherence to the medication and medical regimen [42], recovery [36], psychological problems [41], "coping" [50], fantasised donor-recipient relationships [30], and graft rejection [43]. The three sub-themes that best accommodated all results were *Dynamic psychosocial impact*, *Management*, and *Rejection*.

The whole experience of organ transplantation has a **Dynamic psychosocial impact** that continues after surgery. Recipients reported feeling *Hope*, comparing their *Expectations and reality*, reflecting on the *Donor*, experiencing changes to *Identity*, and continuing to feel a *Sense of inadequacy*.

Participants in eight studies are reported to have described the transplantation as the beginning of *Hope*, enabling them to avoid incipient death and experience a healthier life [27, 29, 32, 33, 36, 44, 45, 51]. They used phrases such as a "second chance", "being born again", "back to normal," and a "resumption of life". Researchers at times characterised these phrases as expressions of gratitude [29, 34, 35, 37, 39, 41, 44] and demonstrations that recipients were "different" people from their pre-transplantation selves [38, 47]. The hope reported in the post-transplant period appeared to be an important component of the dynamic psychosocial impact of the transplant rather than the resource it constituted in the pre-transplant period.

Research participants were reported as comparing their *Expectations* of transplantation with the *Reality* of their experience. Although most participants were satisfied with transplantation, it was found that many experienced a longer than anticipated recovery process and had not expected to encounter problems, including post-transplant illness, various stresses and strains, and the possibility of organ rejection [27, 29, 32, 33, 35, 37, 40, 45, 48, 49]. Participants in an Iranian study expressed regret that they had consented to transplantation because of the resulting suffering, although such feelings were balanced by satisfaction that their lives had been extended [45]; the authors did not report on the time since transplantation of those who were regretful. One paper reported that recipients felt better than expected [36].

The seven studies that included investigations of recipients' reflections on their organ *Donors* found that most avoided or denied thinking about them [30, 35, 38, 39, 41, 45, 46, 48,

50]. Two studies revealed that the language used to convey thoughts about donors was often mechanistic, allowing recipients to separate themselves from the visceral realities of transplant [39, 52]. Those who did disclose such thoughts reported frequently shedding tears and wondering about the age and sex of their donor, whether the donor was a better person than the recipient, and what expectations the donor family had of them; they found it difficult to cope with the realisation that their donor left a grieving family [32, 35, 39, 48, 50, 52]. A few recipients were concerned that they had been complicit in the mutilation of a corpse [39].

Recipients could feel indebted to the donor and the donor's family and regret that saving their own life necessitated another person's death [38, 39, 41, 45]. Recipients were reported as speaking defensively about having been given a scarce resource, some pointing out that they alone were suitable candidates or eager to prove that they were worthy by strict adherence to medical advice [38, 44]. They might have chosen not to reveal their age to the donor's family to avoid provoking regret that the organs were not donated to someone younger. One recipient's account revealed a deep sense of obligation to the donor; recognition of the donated lungs was, at times, the only thing that kept the recipient from committing suicide [37]; the authors did not state how much time had passed since her transplant. Because it was implied in the paper from Iran that recipients had developed relationships with the donors' families [45], we contacted the Organ Procurement Unit at the Shahid Beheshti University of Medical Sciences, Tehran. We learnt that donor anonymity is not mandated in Iran although it is usually practised [personal communication, 29 April 2017].

*Identity* that had previously been defined in relation to a congenitally diseased heart was reported to be uncertain for transplant recipients with some recipients mourning the loss of their original heart [38, 41, 47, 52]. A recipient's relation to donated lungs (and associated sense of identity) could change over time, from perceiving the lungs to be completely external, through a transitional position in which the organs are shared by donor and recipient, to finally accepting the lungs as theirs [30, 38]. Some recipients were reported as musing on the changes in their relationship with the donated organ [52]. These psychological changes were identified as associated with physical recovery [30]. Some researchers found that their participants were either concerned that they would assume the donor's identity or believed that the donated organ had changed their identity [39, 45].

A fluid identity contributed to a *Sense of inadequacy;* an initial release from dependence on others was followed, for some participants, by a realisation that they had not escaped being defined by illness or that they were now a "post-transplant person" [33, 38, 43, 48, 52]. Although some researchers reported that recipients (re)gained a sense of agency, it could be only temporary because they found themselves once again subject to impositions such as taking medication, avoiding certain foods, and exercising [33, 35, 37, 43, 44, 47, 48]. In contrast, other recipients found that mastering these routines gave them a feeling of control and that the process of transplantation revealed an inner strength they had not known they possessed [32, 33, 36, 47, 51]. The vigilance necessary to detect an episode of rejection could contribute to loss of agency, with recipients feeling "married" to the transplant team and restricted by protective measures imposed by families concerned about their health [29, 45]. Gratitude, expressed by recipients towards donors, donors' families, medical staff, and supportive family and friends, could thus be complicated by resentment and guilt, particularly if the outcome were disappointing [27, 30, 33, 35, 37, 41, 44, 47, 51]. A sense of inadequacy could be exacerbated by feelings of disenfranchised grief, arising from recipients' inability to reciprocate "the gift of life" and from unacknowledged or disallowed mourning on behalf of the donors and their families [46, 48].

*Guilt* was a powerful component of the sense of inadequacy. For example, there were indications of magical thinking in which recipients expressed the belief that they could control the

biological process of graft rejection (beyond medication adherence) and would blame them-selves for "causing" an episode of rejection [43]. Transplantation anniversaries were sensitive markers of both celebration and mourning, with some recipients feeling complicit in another's death and those experiencing graft rejection burdened by a sense of having failed to honour an implicit contract to ensure transplant success [30, 33, 37, 41, 44, 46, 47]. Recipients were found to continue experiencing a tenuous grasp on life: organ transplantation does not necessarily resolve the existential tension between life and death [29, 32, 33, 36, 46–48].

Post-transplant *Management* (subdivided into *Medical* and *Support*) encompasses both self-management and external ideas and constraints of what should be managed.

Post-transplant *Medical management* was discussed in 13 papers; recipients spoke of their experiences of the transplant clinic and medical crises as well as the demands of the medical regimen [29, 32, 33, 35, 36, 40, 42–45, 47, 50, 51]. Medical management sometimes merged with self-management because, as is often the case in chronic illness, the patient became part of the management team. The clinic can be a source of fear and anxiety, an intrusive reminder that a trajectory of good, stable health can be interrupted by the results of testing ordered by clinicians [32, 33, 40, 43, 45, 51]. Some participants were concerned about clinics and clini-cians: conflicting advice, dismissive attitudes, and inadequate follow-up [35, 40, 47, 51].

Recipients could find it hard to recognise what symptoms should be reported to clinicians, which sometimes made adherence difficult [29]. It was reported that clinicians emphasised the importance of adherence to a post-transplant regimen that included medication, diet modifi-cation, avoidance of public transportation, and physical exercise [33, 35, 44, 45, 47, 51]. Adher-ence to this regimen could be enhanced by interpersonal relationships, while adherence could be disrupted by unpleasant side-effects, conflicting medical advice, hard-to-follow instruc-tions, "naivete", time constraints, and a reluctance to relinquish favourite foods [29, 35, 40, 42, 44, 47, 51]. Emphasis on adherence could engender such a powerful sense of responsibility that recipients interpreted organ rejection as indicative of non-compliance even when they knew this not to be the case [33, 43, 51].

Recipients were found, in some cases, to have experienced post-procedural trauma, akin to post-traumatic stress disorder, in which events (such as clinic visits, medical procedures to test for rejection or infection, transplant anniversaries, constant vigilance to detect symptoms) aroused the anxiety associated with transplantation surgery, the preceding illness, or rejection [27, 32, 33, 43, 47, 51]. Recipients commonly described being fearful: that they would exhaust all treatment options when faced with graft rejection; of iatrogenic diabetes, weight gain, and hypertension; and the threat of illness and death [29, 33, 36, 40, 43–45, 47, 51]. Fear could be masked by an ostensible carelessness about the consequences of not heeding medical advice [27, 33, 43, 45].

Post-transplant *Support* was found to be both external and internal, including in the two studies that focused on the types and measures of support available to heart and lung trans-plantation candidates and recipients [35, 49]. Sources of *External support* were allied health professionals, transplantation nurses, workplaces, families, social groups [35, 36, 38, 42, 49, 51] and, occasionally, peers [32, 43]. Those who made use of peer support were reported as saying that it was the only source of experiential information about transplantation [43]. There were reports of inadequacies in support from healthcare organisations, workplaces, families, trans-portation systems, and in financial matters [35, 40–42, 45]. Complaints included frustrating failures of information transfer between healthcare organisations, inadequate information about organ transplantation, a lack of understanding about transplantation by employers and colleagues, and inadequate support from family and friends who had been expected to provide it [35, 36, 40, 49, 51].

Post-transplant peer support was commonly given rather than received [27, 33, 43, 44, 47, 51]. Recipients said (or the researchers posited) that they became peer supporters because it had been missing from their pre-transplant experience, to attempt to resolve a quest narrative, to confront death and help others to do the same, as displaced reciprocation for the "gift" of organs, to satisfy the social contract of gift-giving when direct reciprocity is impossible, and for empowerment [27, 33, 43, 44]. It was reported in one paper that, on the whole, heart recipients felt that successful recipients served as role models and that a comparison with fellow recipients could be constructive if the recipient were better off than their comparator [32]. Peer support was not always unambiguously beneficial; giving support could induce guilt in the supporters when those they were supporting became very ill or died, and these adverse outcomes could, in turn, cause anxiety in peer supporters because they emphasised the potential for complications [40, 51].

Recipients who reported on the benefits of returning to work found that external support interacted with personal capacity, yielding feelings of accomplishment, respect, and being a valued member of society [32, 33, 35, 47]. Similarly linking internal and external support, divine intervention was seen to arise from the *Internal support* associated with faith; God was credited with personal survival, the availability of suitable organs, and directing one's life; faith was claimed as engendering a superior coping style [27, 45, 47, 50]. Internal support was evident in the setting and changing of goals, often reprioritising to accommodate better health (or, in some cases, to manage worsening post-transplant health) and in deference to new understanding about the recipient's role in life; goals could also be used as a method of motivation [27, 50, 51]. Recipients could find that a positive attitude led to a perception of control and mastery of post-transplant life and that resilience was strengthened by believing in the potential to recover [32].

Investigating post-transplant ***Organ rejection*** was the primary aim of two studies [29, 43]; the topic was addressed in a further seven [27, 33, 35, 39, 40, 47, 51]. Recipients were reported to be fearful of rejection. The fear was compounded by the difficulty of identifying an episode of rejection because the symptoms were nebulous and variable, leaving some recipients feeling that the post-transplant body was not to be trusted [27, 29, 35, 40, 43, 51]. Diagnostic tests were reported to be emotionally threatening, as was taking immunosuppressant medication to prevent rejection. Some recipients speculated that one cause of rejection was the inherent inequity of the organ donation process, whereas others balanced the threat or experience of rejection against the benefits of a relatively normal life [39, 47], exemplifying the diverse meanings derived from the profound and complex phenomenon of organ donation.

## Discussion

This review is the first to include recipients of hearts, lungs, and hearts and lungs. It was, however, limited by its exclusion of papers not published in English. Its focus on first-hand accounts from recipients might have excluded those who are reliant on a carer for communication. Nevertheless, this review is comprehensive, with great efforts made to locate eligible papers and to be inclusive; it was conducted with rigour.

The review identified 24 papers (from 20 studies) that had used qualitative methods to investigate the experience of being a heart, lung, or heart-lung recipient. The quality of papers included in the review was generally assessed as high. A more detailed and nuanced picture of post-transplantation experience emerges from these papers to complement and enlarge upon quantitative studies that demonstrate the existence of persistent psychosocial distress after transplantation (e.g., [53]). When recipients speak for themselves they describe complex emotions that almost invariably include, along with some distress, satisfaction with being alive and

gratitude to donors and their families. This review has also updated and extended an older review of qualitative research on heart recipients [25]. Our review's findings are consistent with a review of adolescent experiences of organ transplantation [54] apart from adolescents' greater emphasis on the role of parents and siblings.

In the distinct chronological periods of before the transplant, the transplant, and after the transplant, a dynamic psychosocial impact on recipients was evident. The prospect of death presents an existential threat to the self before a donated organ is received and this threat is not banished by transplantation, but replaced by vulnerability to a necessarily compromised immune system and the risk of organ rejection. It can be difficult for those on a waiting list for an organ to understand what life will be like in the post-transplant period. Recipients can be surprised not to be fully healthy and to be still dependent on the clinic. Nevertheless, nearly all recipients were grateful for the extra time given to them by a new heart or lungs, which is consistent with the findings of Seiler et al. [24]. Although it could be expected that the meaning of heart and lung transplantation would change with the passing years, the themes identified are remarkably consistent across all studies.

Recipients endeavoured to sustain internal sources of support, such as hope and faith, but also drew on external sources such as family, friends, the clinic, and peers; this is consistent with the conclusions reached by Conway et al. [25]. External sources of support were not always optimal or even available; this aspect of transplantation care could benefit from further practical implementation, especially the engagement of peers, whether in person or via other means such as online.

Constructs of the self were clearly challenged by receiving an organ from a deceased donor. Challenges came from uncertainty about how another person's heart or lungs could or might change identity; concern about being worthy of such a significant gift, especially when the donor's grieving family loomed large in the imagination; and the continuing dependence on the clinic and on family for daily life and survival. Recipients not only could feel the need to justify that they were worthy of the gift but often also felt guilty: for any way that they might have contributed to needing a donated organ, for hoping that a healthy potential donor would die, for any episodes of feared or actual rejection, and for burdening their families (as has been found for chronic illness in general [55]). The guilt associated with receiving a cadaveric organ distinguishes the potential or actual transplantation recipient from those who do not depend on the donor's death to extend their lives [56].

## Conclusion

The results of this review have implications for the psychosocial care of cardiothoracic transplant recipients. For example, pre-transplant care could be managed to ensure that patients are not left feeling fearful, isolated, and without agency. Given the evidence of a disparity between what recipients expected of life after transplantation and their experience of post-transplant life, further research could usefully investigate why this occurs and how it could be mitigated. For example, do transplantation programs contribute in any way to unrealistic expectations of transplantation outcomes? Is the optimism that can accompany the decision to be placed on a waiting list psychologically necessary to managing the stresses and fears that also accompany the decision? Would it be beneficial to find ways of enabling prospective recipients to learn about the experiences of those who had preceded them? Given the lack of a gender lens in almost all papers, these and other questions could best be answered by considering whether the experience of cardiothoracic transplantation is modified by gender. A systematic review of psychosocial aspects of transplant programs and reviews of the effectiveness of such programs would be a valuable contribution to knowledge and to better patient care.

The main conclusion to be reached by this review is that, despite the profound benefits of receiving a donated heart or lungs—of which recipients are well aware—recipients need continuing psychosocial as well as medical support, based on an understanding of the many complex challenges that confront them.

## Supporting information

**S1 Checklist. PRISMA 2009 checklist.**
(DOC)

**S1 Table. Data & access.**
(DOCX)

## Author Contributions

**Conceptualization:** Claire Stubber, Maggie Kirkman.

**Data curation:** Claire Stubber, Maggie Kirkman.

**Formal analysis:** Claire Stubber, Maggie Kirkman.

**Funding acquisition:** Claire Stubber, Maggie Kirkman.

**Investigation:** Claire Stubber, Maggie Kirkman.

**Methodology:** Claire Stubber, Maggie Kirkman.

**Project administration:** Claire Stubber, Maggie Kirkman.

**Resources:** Claire Stubber, Maggie Kirkman.

**Supervision:** Claire Stubber, Maggie Kirkman.

**Validation:** Claire Stubber, Maggie Kirkman.

**Visualization:** Claire Stubber, Maggie Kirkman.

**Writing – original draft:** Claire Stubber.

**Writing – review & editing:** Claire Stubber, Maggie Kirkman.

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
