## [Decision Letter · Decision Letter 0]

18 Aug 2020

PONE-D-20-03070

The experiences of adult heart, lung, and heart-lung transplantation recipients: A systematic review of qualitative research evidence

PLOS ONE

Dear Dr. Kirkman,

Thank you for submitting your manuscript to PLOS ONE. After careful consideration, we feel that it has merit but does not fully meet PLOS ONE’s publication criteria as it currently stands. Therefore, we invite you to submit a revised version of the manuscript that addresses the points raised during the review process.

We look forward to receiving your revised manuscript.

Kind regards,

Marie-Pascale Pomey

Academic Editor

PLOS ONE

Journal Requirements:

2. Please ensure you have thoroughly discussed any potential limitations of this study within the Discussion section. In addition, please provide the full search strategy for at least one database.

3.In your Data Availability statement, you have not specified where the minimal data set underlying the results described in your manuscript can be found. PLOS defines a study's minimal data set as the underlying data used to reach the conclusions drawn in the manuscript and any additional data required to replicate the reported study findings in their entirety. All PLOS journals require that the minimal data set be made fully available. For more information about our data policy, please see http://journals.plos.org/plosone/s/data-availability.

4.Please include your tables as part of your main manuscript and remove the individual files. Please note that supplementary tables (should remain/ be uploaded) as separate "supporting information" files.

Reviewers' comments:

Reviewer's Responses to Questions

**Comments to the Author**

1. Is the manuscript technically sound, and do the data support the conclusions?

Reviewer #1: Yes

Reviewer #2: Yes

2. Has the statistical analysis been performed appropriately and rigorously? 

Reviewer #1: Yes

Reviewer #2: N/A

3. Have the authors made all data underlying the findings in their manuscript fully available?

Reviewer #1: Yes

Reviewer #2: Yes

4. Is the manuscript presented in an intelligible fashion and written in standard English?

Reviewer #1: Yes

Reviewer #2: Yes

5. Review Comments to the Author

Reviewer #1: This manuscript reports a systematic of qualitative research on the experiences of heart, lung and heart-lung transplantations. One major strength of this paper is the inclusion of a past heart-lung transplant recipient within the research team.

Here are some issues that need to be addressed:

1. Introduction section:

a. P. 4, 2nd paragraph. It is written “The obligation of informed consent requires patients to decide what is best for their own health; in the case of cardiothoracic organ transplantation, patients must determine whether a particular procedure will save their live.” This sentence should be nuanced because most of heart or lung transplant candidates have no other options than transplantation otherwise, the will die.

2. Background section:

a. P.5, 1st paragraph.

i. The authors should provide more details about what they mean when they write that transplant candidates should deal with the stigma associated with organ donation.

ii. The authors should be clearer about what they mean when they mention that transplant recipients could feel guilty arising from ethical questions related to organ donation such as justice, beneficence, utility and harm.

b. The authors should provide some rationale why to include heart, lung and heart-lung transplant recipients in their systematic review. It is unclear for me what this systematic review will bring to the existing reviews.

3. Methods section:

a. P. 7, there is probably a typo when they write the total number of participants.

b. P.9, the authors should mention if they used any software to analyse the data and if a rate of coding agreement was calculated among coders.

4. Results section

a. It would be important to add some citations in order to support the themes and subthemes identified.

b. This section results need to be better presented. We really need to understand what is related to every themes identified by the study team. For instance, p. 10, 3rd paragraph, the authors mention the clinics and their role. However, it is unclear how it is related to the dynamic psychosocial impact. It seems more logically related to resources and support.

c. I suggest that the authors summarizes the themes and excerpts in a Table in order to add clarity in the results presentation.

d. It would also be of interests to know the number of studies which report the different themes identified by the research team.

e. P.17, the authors seem to use interchangeably medical management and self-management which are not the same concept.

f. Peer support is a frequent theme. Maybe the authors should make it as one of the theme.

5. Conclusion section

a. The authors mention that the results have implications for the psychosocial care of heart without providing further details on the implications. I suggest that the authors provide more details on this statement.

Reviewer #2: Thank you for allowing me to review this manuscript that reports findings from a systematic review of qualitative studies on the experiences of adult heart and lung transplant recipients. The authors reported the background and methods in sufficient detail. Strengths were the comprehensive search strategy and comprehensive quality assessment of included studies. The findings were presented with evidence supported to justify the themes.

6. PLOS authors have the option to publish the peer review history of their article (what does this mean?). If published, this will include your full peer review and any attached files.

Reviewer #1: No

Reviewer #2: **Yes: **Aaron Conway

---

## [Author Response · Author response to Decision Letter 0]

7 Sep 2020

We thank the editor and reviewers for their thoughtful reading of our manuscript and their helpful suggestions for improvement. We have specified below our response to each comment and the revisions we have made.

We hope that our manuscript is now acceptable for publication in PLOS ONE.

EDITOR (journal requirements) 

RESPONSE

We have checked our manuscript against PLOS ONE’s style requirements. We think the style was correct apart from our headings. We have corrected our headings, including aligning them left and using bold 18 point font for headings, 16 point for sub-headings and so on. We have not shown the changed headings in track Changes because our names were revealed. However, we think that these changes are obvious.

2. Please ensure you have thoroughly discussed any potential limitations of this study within the Discussion section. In addition, please provide the full search strategy for at least one database.

RESPONSE

All potential limitations of this study are disclosed in the Discussion section of the manuscript. 

The full search strategy is in our protocol. We think it could be sufficient to cite that, but will upload part of the protocol should that be considered necessary.

RESPONSE

Our data are 24 peer-reviewed journal articles. They are all available online. Any restrictions to access are those of the journals’ publishers. Full details are in our reference list. 

We note the response of both reviewers to the data availability question:

“The PLOS Data policy requires authors to make all data underlying the findings described in their manuscript fully available without restriction, with rare exception (please refer to the Data Availability Statement in the manuscript PDF file). The data should be provided as part of the manuscript or its supporting information, or deposited to a public repository. For example, in addition to summary statistics, the data points behind means, medians and variance measures should be available. If there are restrictions on publicly sharing data—e.g. participant privacy or use of data from a third party—those must be specified.

Reviewer #1: Yes

Reviewer #2: Yes”

4. Please include your tables as part of your main manuscript and remove the individual files. Please note that supplementary tables (should remain/ be uploaded) as separate "supporting information" files.

RESPONSE

We have inserted Tables 1, 2, and 3 at the end of the manuscript. 

RESPONSE

We have added the title of our supplementary information file to the end of the manuscript: S1: PRISMA 2009 Checklist. 

REVIEWER 1

6. Introduction section: P. 4, 2nd paragraph. It is written “The obligation of informed consent requires patients to decide what is best for their own health; in the case of cardiothoracic organ transplantation, patients must determine whether a particular procedure will save their live.” This sentence should be nuanced because most of heart or lung transplant candidates have no other options than transplantation otherwise, the will die.

RESPONSE

For optimal results, the transplant may be offered when the candidate is not so sick that they are in imminent danger of death, but not so well that their life expectancy is beyond two years. This is a hard balance for the treating team to strike. People are moved on and off transplant lists when it is considered that the candidate is too sick to survive the surgery or when (for some reason, such as response to a novel treatment), they are considered too well to be a candidate. It is not inevitable that a transplant will save a life or add to life expectancy sufficiently to compensate for the rigours of the surgery and its aftermath. To add the nuance that the reviewer requests, we have amended the sentence as follows: “…must determine not only whether a particular procedure will save their life, but what, if any, improvements they can expect in their quality of life.” 

7. Background section: P.5, 1st paragraph.

i. The authors should provide more details about what they mean when they write that transplant candidates should deal with the stigma associated with organ donation.

RESPONSE

The article we cite goes into more detail than would be appropriate here. We have added examples of stigma. 

8. Background section: P.5, 1st paragraph.

ii. The authors should be clearer about what they mean when they mention that transplant recipients could feel guilty arising from ethical questions related to organ donation such as justice, beneficence, utility and harm.

RESPONSE

We have expanded the sentence to clarify our meaning, giving an example of each ethical aspect.

9. b. The authors should provide some rationale why to include heart, lung and heart-lung transplant recipients in their systematic review. It is unclear for me what this systematic review will bring to the existing reviews.

RESPONSE

Our rationale is stated on page 5: “There has not been a review of qualitative research on recipients of donated hearts and/or lungs; it is therefore time to update the Conway et al. [25] review and to extend it to recipients of donated lungs.” Further, we state in our discussion what has been contributed by our systematic review. One of the main purposes of a systematic review is to combine evidence to provide a more comprehensive view than is possible with individual research publications. It seems to us that this is too familiar to be stated here, although we can do so if that is the editor’s preference.

10. Methods section:

a. P. 7, there is probably a typo when they write the total number of participants.

RESPONSE

We are grateful to have this error pointed out to us. It has been corrected. [

11. b. P.9, the authors should mention if they used any software to analyse the data and if a rate of coding agreement was calculated among coders.

RESPONSE

We have inserted a statement to the effect that no software was used in the analysis. This was a qualitative analysis of qualitative data; inter-rater agreement was therefore inappropriate.]

12. Results section

a. It would be important to add some citations in order to support the themes and subthemes identified.

RESPONSE

If the reviewer is referring to citations of reviewed papers, these appear in Table 1 where the themes identified in each paper are stated. This is a more comprehensive and efficient means of conveying this information than by in-text citations.

13. 

b. This section results need to be better presented. We really need to understand what is related to every themes identified by the study team. For instance, p. 10, 3rd paragraph, the authors mention the clinics and their role. However, it is unclear how it is related to the dynamic psychosocial impact. It seems more logically related to resources and support.

RESPONSE

The themes are interconnected; we have now stated this explicitly on page 9. The participants describe how clinics influence their psychological state and social context. We refer to these interconnections with emphasis on the theme being discussed.

14. 

c. I suggest that the authors summarizes the themes and excerpts in a Table in order to add clarity in the results presentation.

RESPONSE

Table 3, now incorporated in the manuscript rather than as a separate document, gives all the requested information.

15. 

d. It would also be of interests to know the number of studies which report the different themes identified by the research team.

RESPONSE

Table 1, now incorporated in the manuscript rather than as a separate document, gives all the requested information. 

16. 

e. P.17, the authors seem to use interchangeably medical management and self-management which are not the same concept.

RESPONSE

We suggest that, as with much chronic illness, the patient is co-opted into the medical management team, whereupon the line between self and medical management is blurred. We have specified this on page 16.

17.

f. Peer support is a frequent theme. Maybe the authors should make it as one of the theme.

RESPONSE

Peer support was a clear contributor to patient wellbeing in the reviewed papers. As we organised all potential themes into an informative hierarchy, we concluded that peer support could best be represented as a component of other, broader, themes. We suggest that our organised themes are more useful and informative than a far more extensive set of more detailed themes. We argue that we have given peer support the weight it deserved, based on the evidence in the reviewed papers.

18. Conclusion section

a. The authors mention that the results have implications for the psychosocial care of heart without providing further details on the implications. I suggest that the authors provide more details on this statement.

RESPONSE

We have inserted further details. 

REVIEWER 2

No suggestions for improvement.

---

## [Editor Report · Decision Letter 1]

19 Oct 2020

The experiences of adult heart, lung, and heart-lung transplantation recipients: A systematic review of qualitative research evidence

PONE-D-20-03070R1

Dear Pr Kirkman,

We’re pleased to inform you that your manuscript has been judged scientifically suitable for publication and will be formally accepted for publication once it meets all outstanding technical requirements.

Kind regards,

Marie-Pascale Pomey

Academic Editor

PLOS ONE
---

## [Editor Report · Acceptance letter]

28 Oct 2020

PONE-D-20-03070R1 

The experiences of adult heart, lung, and heart-lung transplantation recipients:A systematic review of qualitative research evidence 

Dear Dr. Kirkman:

I'm pleased to inform you that your manuscript has been deemed suitable for publication in PLOS ONE. Congratulations! Your manuscript is now with our production department. 

Kind regards, 

on behalf of

Dr. Marie-Pascale Pomey 

Academic Editor

PLOS ONE